



# Technical note on incorporating natural variability in master recession curves

Thomas A McMahon[1], Rory J Nathan[1], Richard George[2]

[1] Department of Infrastructure Engineering, The University of Melbourne, Parkville, 3010, Victoria, Australia
[2] Department of Primary Industries and Regional Development, PO Box 1231, Bunbury, 6231, WA, Australia

*Correspondence to*: Rory Nathan (rory.nathan@unimelb.edu.au)

**Abstract.** In this technical note, we hypothesise that the master recession curve (MRC) is a continuum rather than a single average curve and the natural variability as evidenced in the range of MRCs represents aleatory uncertainty across the continuum and is the result of antecedent hydroclimatic conditions and heterogenous storage conditions in the unconfined aquifer/s feeding the streamflow. For four streams, representing the range of Australian hydrology, master recession curves

were computed for five aleatory conditions (90, 75, 50, 25 and 10 percentiles) using the correlation technique. Observed recessions were superimposed on the plots confirming that the continuum of MRCs represented the observed conditions. For one stream, the Northern Arthur River (a 437 km$^2$ in Western Australia yielding 2.7 mm runoff per year), a qualitative model based on field observations supports the continuum concept.

## 1 Introduction

This technical note describes for periods of no rainfall a comparison between daily master recession curves (MRCs) of a stream, computed using the correlation approach (Langbein, 1938; Federer, 1973; Boughton, 2015), and observed daily streamflow recessions for four catchments representative of the Australian hydrologic landscape. MRCs, which are plots of discharge against time, are unique representations of streams, their catchments and their associated perched or deeper aquifer systems. The shape of an MRC depends on the unconfined aquifer/s and regional groundwaters that feed the stream. The shape

is also affected by transmission losses in the stream which include evaporation from the stream water surface, evapotranspiration from the directly connected riparian zone, and from the stream as seepage to the bed, and at high flows recharge to the banks (McMahon and Nathan, 2021). In this paper, interflow, which is rainfall that on infiltrating moves laterally through the upper soil often returning to the surface downslope prior to joining the stream, is considered part of the MRC. However, any rainfall that becomes surface runoff and remains so is not considered part of the baseflow recession

process where baseflow is the combination of groundwater and delayed sub-surface flows (Shaw, 1994).

Master recession curve analyses have been part of the hydrologic tool kit for the past ninety years. The master recession curve was known initially as a composite recession curve (Linsley et al., 1958, p150) or a normal recession curve (Chow et al., 1998, p134). According to Chow et al. (1988, p132), Horton (1933) was the first person to describe the normal depletion curve (or

master baseflow recession curve as noted by Chow et al., (1988, p134)). The literature on MRCs is extensive and covers





reviews, procedures to estimate MRCs, and discussions of their uses. The major reviews include Toebes and Strang (1964), Hall (1968), Tallaksen (1995), Smakhtin (2001) and Brodie and Hostetler (2005).

Literatures describing or developing procedures to estimate MRCs include: Nathan and McMahon (1990) (applied correlation and matching strip methods); Arnold et al. (1995) (estimated slope of baseflow recession); Raaii (1995) (compared five techniques to estimate MRCs);  Lamb and Beven (1997) (used saturated zone data); Sujono et al. (2004) (applied Wavelets); Mizumura (2005) (applied Kinematic and Diffusion Wave models); Millares et al. (2009) (compared upward and downward fragments to estimate MRC); Griffiths and McKerchar (2010) (developed an equation for MRC); Gregor and Malik (2012) (applied Genetic algorithm); Fiorotto and Caroni (2013) (considered a statistical interpretation); Boughton  (2015) (estimated groundwater storage and transmission loss); French (2015) (suggested maximum recession rather than average); Nimmo and Perkins (2018) (examined episodic MRC); Carlotto and Chaffe (2019) (automated MRC); Duncan (2019) (compared procedures); Singh and Griffiths (2021) (applied to ungauged catchments); Whitaker et al. (2022) (developed spreadsheet procedure for matching strip method); Kim et al, (2023) (examined MRC as attractor in machine learning); Latuamuty et al. (2024) (tested MRC equations); and, Margreth et al. (2024) (compared several methods).

Applications of MRCs are discussed by: Brown (1965) (woodland watersheds and seasonal effects); Nathan and McMahon (1990) (assessing matching strip and correlation techniques); Boughton (1995) (defining baseflow storage); Rivera-Ramirez et al. (2002) (estimating critical level of low flows); Chapman (2003) (estimating losses by seepage and /or evaporation); Somorowska et al. (2004) (estimating groundwater storage); Berhail et al. (2012) (low flow separation); Chen et al. (2012) (estimating hydraulic parameters in Karst aquifer); O'Brien et al. (2014) (comparison with alternative approaches); Kavousi and Raeisi (2015) (residence time in Karst aquifers); Griffiths and McKerchar (2015) (predicting streamflow recession); Ambroise (2016) (hypothesis testing and water balance studies); Yang et al. (2019) (seasonal MRCs); Parchami et al. (2024) (assessing spatial and temporal changes in MRC); and, Trotter et al. (2024) (catchment  storage and drought).

Here we propose an approach to characterise the natural variability inherent in the rates of streamflow depletion in the absence of rainfall (i.e., in MRCs). We describe the method's application to four catchments representative of a wide range of hydroclimatology in Australia and compare the derived MRCs to observed recessions.

## 2 Adopted approach

### 2.1 Development of Master Recession Curve

Traditionally, an MRC is a plot of daily streamflow (logarithmic scale) as a function of time as in Fig. 1. From A to B, the MRC is upwardly concave until the flow reaches the maximum recession constant (B) after which the logarithm of flow decreases linearly (B to C). The difference in the curves (B - D) and (B - C) is transmission loss.





The correlation method is adopted herein to compute an MRC. It entails binning the daily recession constants ($K_j = Q_j/Q_{(j-1)}$,

for $0 < K_j < 1$, where $Q_j$ is the daily streamflow at time j using constant bin sizes varying between 50 and 200 items. Our method

follows Boughton (2015) although he used varying bin sizes. In addition to the above limits, values of K are also excluded if

they are potentially affected by rainfall. The number of raindays to be eliminated prior to plotting the recessions is a function

of catchment area. We base the number of raindays on Linsley et al. (1982, Equation 7-4).

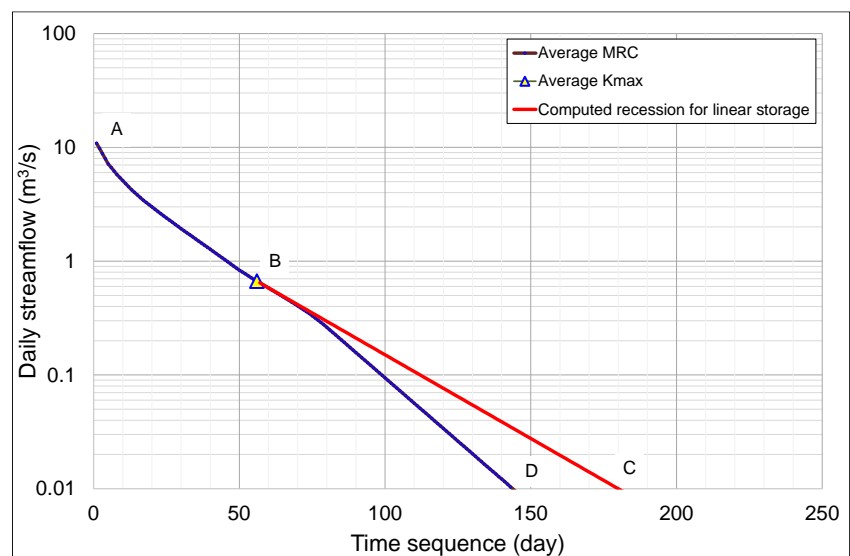


**Figure 1 Example of an average Master Recession Curve (MRC) for Gibbo River at Gibbo Park (401207), Australia including baseflow for a linear model and the average maximum recession coefficient (Kmax). ABC is a traditional average MRC plot whereas ABD is an average MRC incorporating the effect of transmission loss.**

Traditionally, an MRC represents the average rate of decline in streamflow following isolated periods of rainfall after surface

runoff contributions have ceased. The average rate of decline is determined from the analysis of many individual recessions,

whose shape at any point in time depend on the initial water content of the unconfined aquifer and other sub-surface processes

that together feed the stream. Also, as noted above, transmission losses affect recession shape. The initial water content of the

aquifer/s and sub-storage systems is a result of the preceding climate and aquifer antecedent storage as observed by Bart and

Hope (2014). In this analysis we acknowledge that while the MRC is traditionally recognised as representing the average rate

of decline, it is sensible to characterise the natural variability (or aleatory uncertainty) in the MRC that arises from variable

antecedent hydroclimatic conditions and heterogenous storage conditions in the aquifer. Hence, in contrast to others (Tallaksen,

1995; Duncan, 2019), our concept of an MRC of a stream is that it is a continuum of curves that results from the initial water

content of the sub-storages feeding the stream and from the spatial heterogeneity in their physical configuration. These storages

range from being full following a very wet period to a near empty condition following a long dry period. In our analysis, we





represent these two extreme conditions by MRCs for 90%'ile and 10%'ile non-exceedance frequency curves respectively. Three intermediate conditions are also included: 75%'ile represents a moderate wet period whereas 25%'ile is for a moderate dry period. The 'average' initial condition is represented by a 50%'ile MRC. Here, we express variability in terms of an exceedance probability rather than in terms of a physical variable, for example, antecedent rainfall. That is, in some catchments

it might be reasonable to expect that an individual recession might follow an MRC that is exceeded 30% of the time, and another might follow an MRC that is exceeded 70% of the time. Our approach is not inconsistent with Fiorotto and Caroni (2013) who described an MRC in terms of a stochastic process rate than a deterministic one.

It is rare for observed daily streamflows to be available for long periods without the intrusion of rainfall or observational

tolerance issues, or measurement malfunction, and so plots of actual recessions (streamflow volume Q vs time t) rarely show a complete shape of the recession. For example, Figure 2 of Yeh and Huang (2019) illustrates these points. For their 1992 data, there is no recession for days 14 and 15, nor for days 16 and 17 as the estimated flows are equal within each pair. Furthermore, presumably because of rainfall, the recessions do not extend beyond day 17 (1992 data) and day 13 (2012 data). Virtually all investigators are confronted with these problems.


In the proposed approach, various exceedance percentile values for the MRC are estimated from the binned daily Kj and Qj pairs. For each constant sized bin, acceptable recessions, as defined above, are adopted in which the 10%,'ile, 25%'ile, …, and 90%'ile recession constants are computed for each bin. An example is Fig. 2. To provide for varying initial conditions according to catchment wetness (and hence the initial averaged state of the unconfined aquifer storage), the 10%'ile, 25%'ile,

…, and 90%'ile daily Q values are also estimated. Thus, the initial 10%'ile discharge is associated with the 10%'ile MRC, and so on.


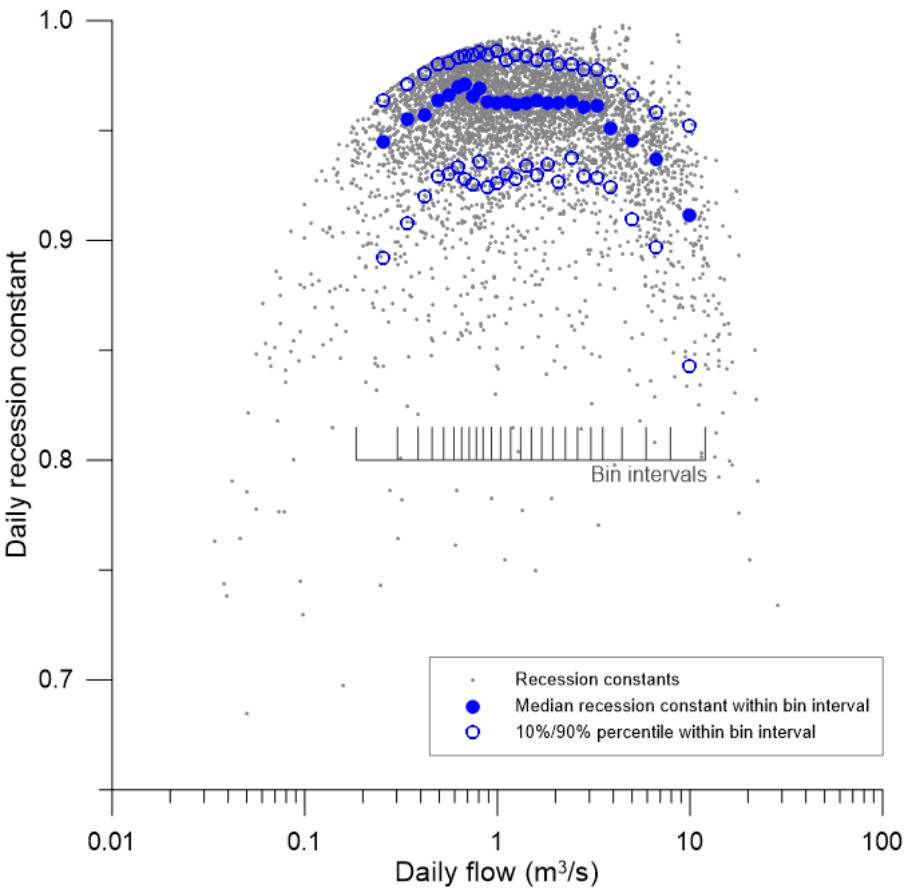

**Figure 2 Relationship between daily recession constant and daily flow for Gibbo River at Gibbo Park (401217) showing median, 10%ile, and 90%ile recession constants in each bin of 200 values.**

## 2.2 Application guidelines

In developing daily MRCs, we adopted the following guidelines:

1. Choose at least ten consecutive decreasing daily discharges that are for days without rainfall; the first x days are not used in the constructing the MRC in an attempt to reduce the potential for surface runoff to affect the recession. In our analysis, 'x' days is defined by Linsley et al. (1982, Equation 7-4).

2. Acceptable time-series, plotted on a log Q (ordinate) versus time (abscissa) graph, should be gently concave (representing non-linearity, prior to the maximum recession constant), convex (subsequent to maximum recession constant i.e., affected by transmission loss), or a straight line from the maximum recession constant i.e., the discharge is from a linear storage system not exhibiting transmission loss. This latter condition is most unlikely as all streams will have





evaporation/evapotranspiration loss, although in some cases the loss may be too small to affect the daily discharge estimates.

3.  In cases where discharges diverge upward from the main curve, these are deleted. Such diversions could occur at either end of the plotted time-series. The earlier values may be the result of some residual surface runoff not being removed by

125       waiting x days before the plot begins. Or the last value (or last few values) could be the result of catchment rainfall (and, therefore, runoff) not identified in the observed rainfall data.

4.  The resulting plotted data should have at least seven consecutive recessive data points i.e., the recession does not include days with equal discharge. This seven-day limit is subjective and there is a trade-off between sufficient data points to define a recession and sufficient recessions to define the master recession curve. (In the example applications below, we adopted

130       a minimum of seven days.)

### 2.3 Study catchments

The comparison between the computed MRCs and observed recessions was made for four streams as set out in Table 1. They cover the hydrology across Australia as defined by mean annual catchment runoff from about 3 to 2000 mm/year.

As noted above, the number of raindays to be eliminated prior to plotting the recessions is a function of catchment area and for Gibbo, Northern Arthur and South Johnstone Rivers three days were eliminated whereas for Myall River the initial four days of recessions were eliminated.

**Table 1 Attributes of study catchments and details of plotted recessions**

| Gauging station | Station reference no. | Latitude, Longitude | Köppen class# | Area (km²) | Data | Mean annual runoff (mm/year) | Annual Cv* | No. of plausible recessions ≥ 7 days | No. of plotted recessions |
|---|---|---|---|---|---|---|---|---|---|
| Northern Arthur River @ Lake Toolibin Inflow | 609010 | 32.905ºS, 117.614ºE | Csa | 437 | Sep '78 – Feb '22 | 2.7 | 2.6 | 24 | 22 |
| Myall Creek @ Molroy | 418017 | 29.799ºS, 150.583ºE | Cfa | 865 | Dec '78- Feb '22 | 37 | 1.28 | 63 | 55 |
| Gibbo River @ Gibbo Park | 401217 | 36.756ºS, 147.709ºE | Cfb | 390 | Aug '71 – Feb '22 | 283 | 0.53 | 65 | 64 |
| South Johnstone River @ Upstream Central Mill | 112101B | 17.609ºS, 145.979ºE | Af | 398 | Nov '74 – Feb '22 | 2027 | 0.38 | 32 | 30 |

# Csa: temperate, dry hot summer; Cfa: temperate, no dry season, hot summer; Cfb temperate, no dry season, warm summer; Af: tropical, rainforest (Peel et al., 2007)
    * Cv is coefficient of variation of annual streamflows





## 3 Results


For each stream, MRCs were computed for five non-exceedance frequencies: 10%, 25%, 50%, 75% and 90%'ile values, representing very dry initial conditions (10%'ile) to very wet initial conditions (90%'ile). The resulting MRCs obtained for Gibbo River at Gibbo Park are shown in Fig. 3 in which the observed recessions are superimposed on the computed MRCs for the five frequency values. Also superimposed on this figure are the maximum recession constants for each condition.

Equivalent plots for other sites are provided in supplementary material.

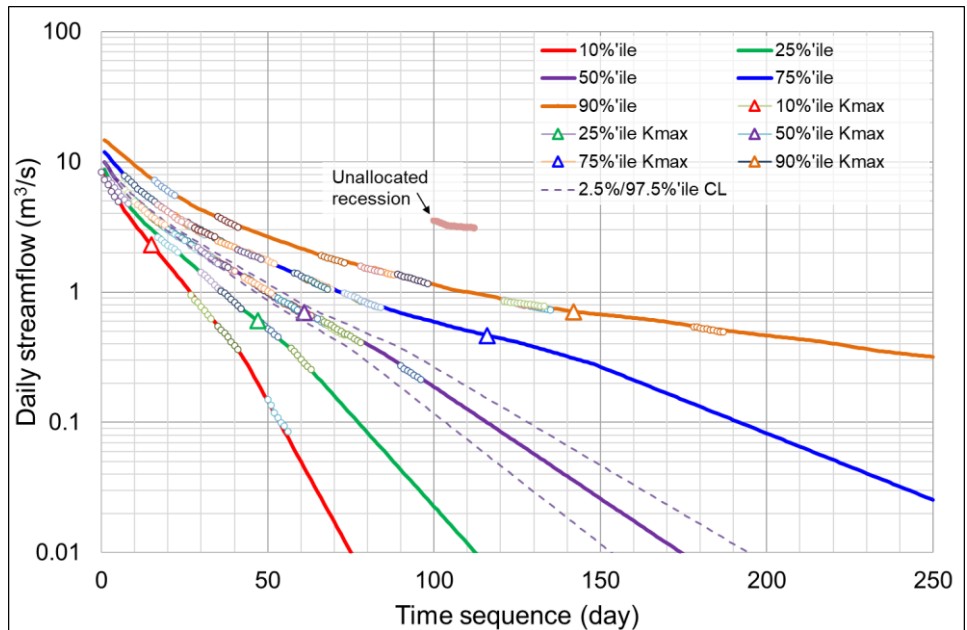

**Figure 3 Comparison of daily MRCs computed using a constant bin correlation method, with observed recessions during rainless periods for Gibbo River at Gibbo Park (401217). Three rainless days were required before data**

**were acceptable in the analysis. The modelled recessions are for five percentile values – 10%, 25%, 50%, 75% and 90%. Maximum recession constants are located in the figure. The 2.5% and 95.5% confidence limits (CL) for the 50% MRC are also included. One acceptable recession was not allocated to one of the five MRCs.**

The number of plausible recessions plotted in the figures are listed in the last two columns of Table 1. The numbers varied

from 24 for Northern Arthur to 65 for Gibbo whereas the final numbers plotted and allocated to a specific percentile MRC curve varied from 22 for Northern Arthur to 64 for Gibbo.





To assess the robustness of manual allocation of the observed daily recessions, for each recession in the Northern Arthur data (Fig. S3) the value of the observed daily recession was compared to the modelled recession curve. Two metrics were computed

based on the observed and modelled recessions: the standard correlation coefficient and the Nash-Sutcliffe Efficiency (Nash and Sutcliffe, 1970). The median correlation coefficient was 0.995 and the median Nash-Sutcliffe efficiency was 0.962, where the differences between the minimum and maximum of these statistics were 0.015 and 0.315, respectively. Overall, the values for both metrics are high, and confirms the adequacy of the manual allocation. However, it was observed that the lowest Nash-Sutcliffe value (NSE = 0.667) and another slightly higher value could be increased by shifting the observed recessions by one

time step.

Figure 4 considers the influence of wet and dry periods on the plotted data with reference to Fig. 3 for Gibbo River. The plots were compiled for the wet and the dry 6-months periods. The two periods were based on the mean monthly streamflows for Gibbo River at Gibbo Park in which the average streamflow during the wet period from July to December was 3.9 times the

average streamflow during the dry period, January to June.

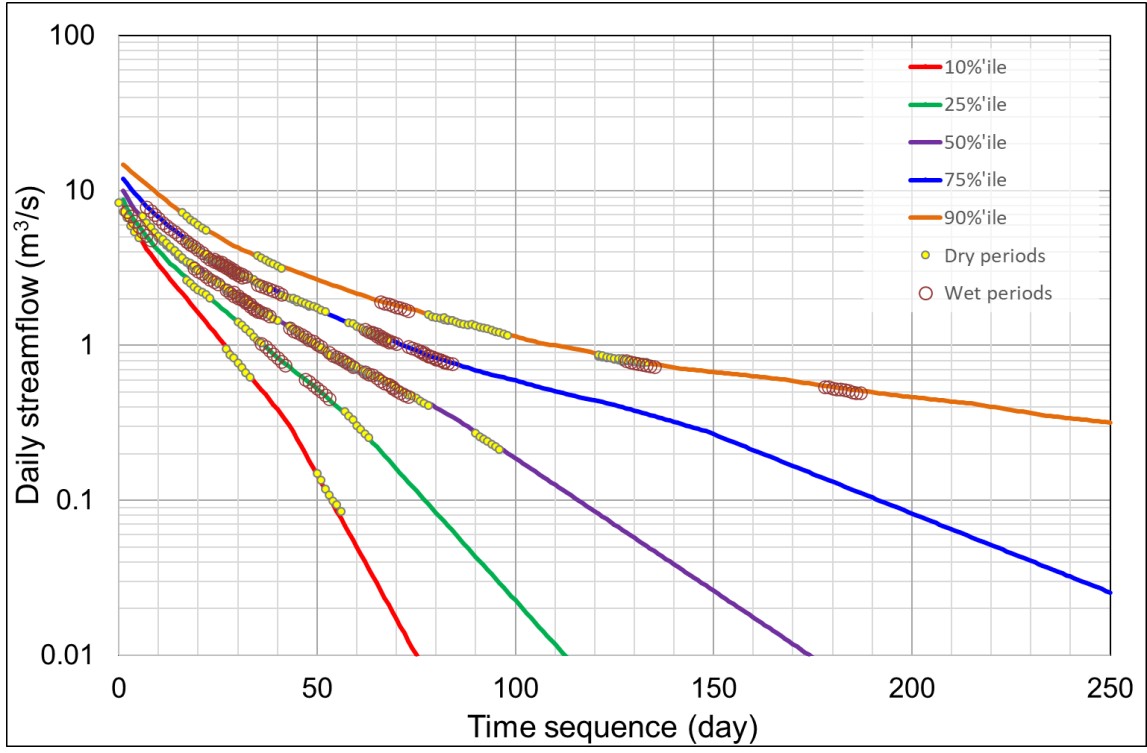

**Figure 4 Comparison of observed daily recessions for wet and dry periods separately superimposed on the estimated MRCs for Gibbo River at Gibbo Park (401217).**






## 4 Discussion

### 4.1 Representation of natural variability

The observed recessions for all four streams (shown in Figs. 3 and 4, and in Supplementary Material, a total of 169 across the four streams) plot neatly on the computed MRCs. This is expected, of course, as both sets of results are drawn from the same

data. Furthermore, the range of behaviour between the 10%'ile to 90%'ile MRCs provides much scope to ensure the observed recessions fit reasonably well to one of the five curves. The plots for Myall River (Figure S2) is an exception where eight recessions were not allocated to one of the five MRCs. Most would have plotted on intermediate MRCs.

Two plots are provided for Northern Arthur. Figure 5(a) shows the recession data for the winter to mid-spring period whereas

Fig. 5(b) is for the summer to early autumn period. These two plots are discussed in detail later.  Noting that at the Lake Toolibin stream gauging station for Northern Arthur River, there are, on average, only 43 days of flow per year and so the number of plausible recessions is limited. It is also noted that the minimum flow plotted in these figures is about five times larger than the minimum that can be estimated under field conditions (McMahon and Peel, 2019). It is comforting to note that for this stream the observed recessions in Fig. 5(b) are, overall, consistent with the computed MRCs. The episodic nature of

the Northern Arthur River hydrology is evident in Fig. 5 where the two of the three unallocated recessions are for climate conditions more extreme than the 10 - 90%'ile plotted range.

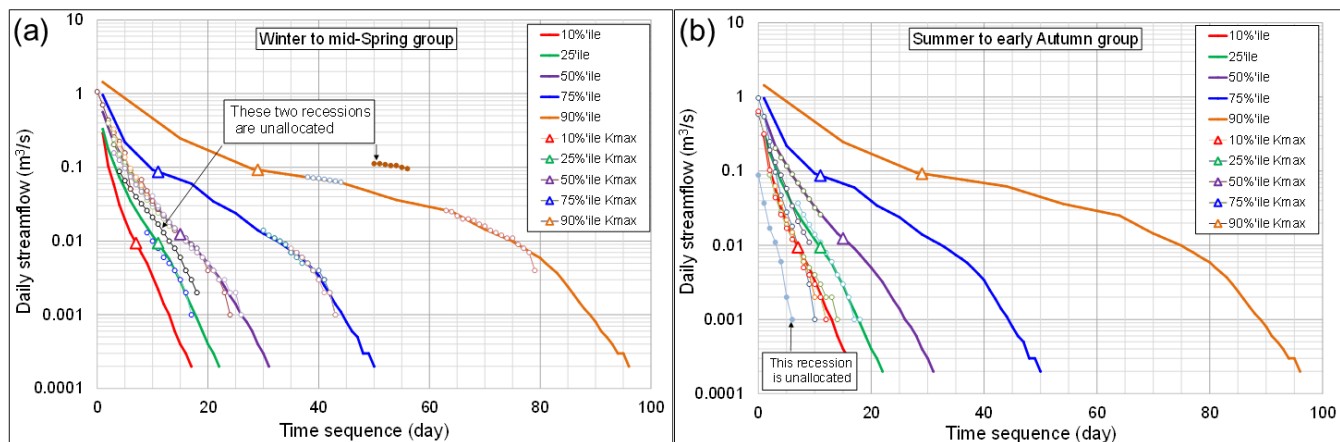

**Figure 5 Comparison of seasonal daily MRCs for Northern Arthur River at Lake Toolibin Inflow (609010) with observed recessions**

**during rainless periods for (a) winter to mid-spring and (b) summer to early autumn seasons. Three rainless days were required before data were acceptable in the analysis. The modelled recessions are for five percentile values – 10%, 25%, 50%, 75% and 90%. Maximum recession constants are shown in the figure. Two acceptable recessions were not allocated to one of the five MRCs.**





We have not found plots similar to Figure 3-5 in the literature comparing observed recessions with probabilistic curves, although Kienzle (2006) identified two very different MRCs for a 7675 km$^2$ watershed in Alberta, Canada. Fiorotto and Caroni (2013, Figure 6) incorporates a range of recessions as a function of probability by interpreting the master recession in terms of a stochastic process. On the other hand, Yang et al. (2019) developed separate MRCs for four flow regime conditions, high, moist, low, and dry and, separately, for the four climate seasons. The approach by Gao et al. (2023), inter alia, is similar to our

method but they do not map the developed recessions to the observed data. Furthermore, their Figure 4(a) suggests that their adopted recessions are not monotonic upwardly concave, and the data appear to be significantly influenced by rainfall during the recessions.

Our plots confirm that the adopted correlation method to estimating MRCs is a valid alternative to other approaches and is

easily adapted to characterising natural variability and uncertainty. More importantly, however, is the observation that measured recessions were found for all four streams that exceeded their respective maximum recession constants, even for the extremely wet South Johnstone with a mean annual runoff of more than 2000 mm/year. This observation independently supports Boughton (2015), who found transmission losses for all but two of one hundred streams covering eastern humid Australia that he analysed. We highlight this point as most international journal papers dealing with classical recession analysis

pay little attention to this important feature of an MRC.

Analysis of recessions over wet and dry periods for Gibbo River (Fig. 4) shows that the observed wet period recessions extend across the five computed MRCs whereas the observable recessions for the dry period cover mainly the 25, 50, 75 and 90%'ile MRCs. This distribution across the two periods is observed despite the fact that the dry 6-month period yields nearly one

quarter the runoff generated during the wet period. The figure is important in that it demonstrates the difficulty in classifying catchment conditions in terms of general wet and dry periods, probably because we are examining the behaviour of individual events. The fact that the sets of derived curves and the observed recessions are near identical is reassuring in that it demonstrates that the processes are not easily treated as lumped averages, but rather as heterogenous units.

Another point to note relates to the confidence with which we can estimate the different percentile MRC curves. As pointed out earlier, we are suggesting the different percentile MRC curves represents the natural variability due to initial climate conditions and the configuration of the aquifers feeding the stream; in other words, the source of variability is the aleatory uncertainty due to antecedent conditions and its influence on a catchment with spatially heterogenous physical properties. However, our ability to identify the different percentile MRCs is limited by the period of available observations, where the

uncertainty around the derived MRCs decreases as the sample size increases. This source of uncertainty represents the epistemic uncertainty due to sampling variability, and this can be readily characterised by using non-parametric bootstrapping. To this end, the uncertainty around a given percentile MRC is characterised by resampling (with replacement) estimates of the recession constant within each bin. An example of the epistemic uncertainty around the median MRC for Gibbo River is shown





by the (dashed) confidence lines in Fig. 3. For the available data, it is seen that the epistemic uncertainty is much smaller than
the aleatory uncertainty.

## 4.2 Hypothesis based on inductive reasoning

But what is the explanation for the variability exhibited in the Figures 3 – 5? Our proposed hypothesis is that, in the catchments
we are considering, there are many sub-catchments (they have Strahler stream orders greater than three (Strahler, 1957)) with
unconfined aquifers and delayed sub-surface flows. These sub-surface storages vary greatly in terms of dimensions, hydraulic
properties, location and elevation relative to the nearest stream. This heterogeneity within a catchment combined with a varying
climate is hypothesized to produce a continuum of MRCs as illustrated in the Figures 3 – 5.

Because Northern Arthur is, hydrologically speaking, a well-instrumented and researched catchment (Callow et al., 2008,
2020), it is used to assess a qualitative model explaining the observed variability (aleatory uncertainty) in the MRCs (Fig. 5;
see also Plates S1 and S2 in supplementary material which show respectively the mid-western catchment and the stream
gauging station location). Two periods of activity are identified, with the number of recessions over the winter to mid-spring
period being twice that over summer to early autumn (Figure S6, supplementary material). Consider Figs. 5(a) and 5(b) which
show the recessions for Northern Arthur River. We deduce that the very steep 10%'ile MRC that occurs over about 15 days
following rain on a very dry catchment occurs from an unconfined aquifer (denoted as A) with very high hydraulic
conductivity. On the other hand, the considerably flatter 90%'ile MRC occurs after a very wet period and is likely fed from an
aquifer (denoted as B) that exhibits a much more attenuated response to rainfall due to having either larger storage or lower
hydraulic conductivity. For simplicity we may assume that these two aquifers, A and B, are the only groundwater sources
discharging to the stream. After a very wet period that fills both aquifers, aquifer A will empty very quickly while aquifer B
will dominate the recession. However, if aquifer A, the less attenuated aquifer, is located in the catchment such that after a
long dry period there is sufficient rain to replenish it yet insufficient for aquifer B to discharge, a steep recession will result.
This explanation must be tempered by noting that stream evaporation transmission losses probably will be high during the
summer to early autumn periods (average area potential evapotranspiration (AAPE) ~5 mm/day) but much less during the
winter to mid-spring periods (AAPE ~2.5 mm/day) (Wang et al., 2001).

The above simple qualitative two-stage model of the surface water-groundwater connectivity proposed for the Northern Arthur
catchment is not inconsistent with the three-stage surface water-groundwater connectivity and vertical recharge processes
described by Callow et al. (2020). Their description is based on intensive instrumentation of the catchment and detailed
analyses. As described by Callow et al, (2020), during the winter period soils saturate and macropores close, matrix flow
dominates, and surficial aquifers become connected; although a vertical hydraulic gradient continues, the system acts as a

semi-confined aquifer, and as the aquifers connect, a transition to the bottom-up groundwater discharge occurs. This description applies to the winter to mid-spring period resulting in the observed recessions depicted in Figure 5(a).

From late spring, groundwater levels fall and, as the system dries, de-coupling between the surface and groundwater occurs, and macropores re-develop. Valley floor areas dry with large surface cracks. Top-down recharge occurs as a result of high

infiltration facilitated by the macropores and surface flows. Observed recessions in Figure 5(b) are the result of these processes. In their discussion of the surface-subsurface processes involved, Callow et al. (2020) describe the transitions between the two main stages of low flow generation in Northern Arthur catchment. Referring to Figure 5, we surmise that some of the intermediate recessions (75%, 50% and 25% 'iles) represent these transitions as the system passes from a wet period to a dry period and back again to the next wet phase.

**5 Conclusion**

This comparison between computed master recession curves (MRCs) and observed recessions for rainless periods confirms the usefulness of the computed MRC constructed using correlation between adjacent days. We hypothesise that the master recession curve is a continuum rather than a single average MRC and the variability across the continuum is the result of variable antecedent conditions in the unconfined aquifers and other sub-surface storages, their relative location, and the

heterogeneity of their hydraulic properties. Our proposed approach is consistent with field data from the Northern Arthur catchment. The master recession curves examined herein support the notion that MRCs should be examined beyond the maximum recession constant to provide a more complete picture of the MRC than is portrayed in the published literature.

**Acknowledgments**

Daily discharge data for this project were from the Australian Bureau of Meteorology's (BOM) Hydrologic Reference Station

project website http://www.bom.gov.au/water/hrs/index.shtml.

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
