# Peer review of "Technical note on incorporating natural variability in master recession curves"

_Hydrology and Earth System Sciences, 2024_

## Author Comment (AC1)

**Response to Anonymous Referee #1**

1. The paper focus on incorporating natural variability in master recession curves. It is of great significance for recession analysis. The topic has received much attention in recent years. However, there are still many key issues that need to clarify.

Response: First, we thank the reviewer for their comments. As far as we are aware, there has been little attention in recent years in understanding the natural variability in traditional master recession curves (MRC) ( $Q \sim t$ ) which is the subject of our paper. Of the 30 odd papers listed in our review of MRCs (Lines 34 to 54), several address seasonality but not variability per se. Regarding recession slope analysis ( $dQ/dt\sim Q$ ), virtually, all attention has been directed towards understanding and applying recession slope analysis following Brutsaert and Nieber's (1977) (BN77) pioneering work in 1977. We agree with the Reviewer that there are many issues that still need to be clarified in adopting BN77 and variations to date. However, this paper concentrates on the application of the traditional recession analysis ( $Q \sim t$ ) to better understand and characterise low flow hydrology.

2. Firstly, there has been a great development of methods for streamflow recession analysis, and there are many alternatives to the correlation method, so why not consider other more popular and sophisticated methods, such as recession analysis based on -dQ/dt~Q. Response: There are two approaches to streamflow recession analysis. First, there is the traditional recession analysis grounded in Horton's (1933) paper where he proposed the concept of a master recession curve (plotting daily discharge against time in days). This approach has been used for many analyses (forecasting low flows Hall (1968); storm hydrograph recessions Tallaksen (1995); estimating transmission losses Boughton (2015)). Second, there is the recession slope analysis (dQ/dt~Q) of Brutsaert and Niebler (1977) and its variations over the past 45 years.

In our recent paper, McMahon and Nathan (2025) (which became available after we submitted our original manuscript) we noted these variations, and proposed the traditional approach which is grounded in the  $Q \sim t$  function rather than its derivative as adopted by Brutsaert and Nieber (1977) to estimate aquifer hydraulic properties. Our method therein is based on the traditional MRC and utilizes the maximum recession constant thus allowing the characterisation of aleatory uncertainty due to natural climatic variability, which is still a largely unaddressed issue in  $-dQ/dt \sim Q$  analyses.

**We propose to add an additional paragraph to the end of the Introduction (Section 1) that would clarify our objectives.**

3. Secondly, the correlation method faces challenges on applications, such as the observation noise in streamflow bring large uncertainty for the calculation of K, especially during low flow period, the inability to consider the continuous recession process by mainly only using the information from the adjacent days.

Response regarding noise in the estimation of *K*: There are several points of importance here.

(a) Some low flow analyses using daily discharges are affected by data imprecision. We are aware of methods to deal with this (see McMahon and Nathan, 2025) where we apply Rupp et al (2009, Appendix A).

- (b) Importantly, we note that the maximum recession constant estimated in our analysis is at a discharge often an order of magnitude larger than the very low discharges where imprecision occurs.
- (c) In the past, traditional recession analysis (Q ~ t) has been applied to daily discharges without adjusting the extremely low discharges for data imprecision. Thus, we have followed this approach in our comparisons in Figures 3 to 5 and Supplementary Figures S1 to S5. In terms of observed recessions as plotted in these figures, few discharges are in the range where data imprecision is an issue.
- (d) Applying a data imprecision adjustment (e.g., Rupp et al (2009) as in McMahon and Nathan (2025)) to the daily discharge data used to produce the master recession curves would be inappropriate because we are comparing the computed MRC with observed daily recessions. We will modify Section 2.1 of the manuscript to reflect this discussion.

Response regarding inability to consider continuous recessions: One can use continuous observed daily discharges to develop a master recession curve but with considerable difficulty. Our landmark analysis of 180 Australian streams (Nathan and McMahon, 1990) did exactly that and we documented those difficulties and benefits. Unfortunately, the procedure utilizes only a small fraction of the total daily record (and the fitting process is subject to considerable subjectivity). For example, for the Gibbo River data used herein, there were 5092 pairs of *K* and *Q* to establish the *K* distribution that define the percentiles but only 527 days are available to plot the recessions. We are confident that the correlation technique is an appropriate method to estimate a master recession curve within the  $Q \sim t$  framework. We see no need to amend the manuscript.

4. Further, the appropriateness of generating MRCs with frequency from the percentiles of the distribution of K has not been adequately demonstrated. Thus, the robustness of the correlation method and the rationality on computed MRCs at different frequency are needed. Response: When we submitted this technical note our paper, the McMahon and Nathan (2025) paper had not been accepted so we were reluctant to offer it as an example of generating MRCs from percentiles of the distribution of *K*. As far as we are aware Gao et al (2023) and McMahon and Nathan (2025) and are the only authors to characterise the influence of natural variability on recession behaviour, and the current manuscript is the only paper to do this using the Q ~ t approach. In a revised manuscript (at about Line 105) we will expand our explanation of the method and include mention of the McMahon and Nathan (2025) reference.

**Detail comments:**

1. Lines 11-13. How about the other three catchments?

Response: We will modify the sentence to make it clear that there are field data for Northern Arthur but not for the other three catchments. Northern Arthur was suitable for detailed study. The Abstract will be modified to reflect this point.

2. The section of Introduction. It is hard to catch the key points.

Response: Yes. We see the need to expand the last paragraph of the Introductory Section. This discusses natural variability in the recession process and leads to the approach adopted in the paper.

- Lines 34-54. This paragraph suggests a tabular presentation. More introduction on the
  research of the natural variability of recession processes is suggested.
  Response: Although our manuscript is written as a Technical Note, if there is an opportunity
  we will expand and recast the tabular material into text format with some concluding
  comments.
- 4. Line 64. How to use the correlation method to compute MRC is not clear. More details are needed.

Response: We thank the reviewer for identifying this inadequacy. We will add an introductory paragraph at the start of Section 2 which identifies the two steps in our approach. We will also change the heading of Section 2.2 and the first sentence to represent more clearly step two of our analysis. Also, more details will be provided on the steps to determine an MRC using the correlation procedure.

5. Figure 1. More evidences or refences are needed to justify kmax.

Response: The maximum recession constant has been used by many researchers under different guises. It is related to the characteristic time (Brutsaert and Lopez, 1988), or to residence time (Chapman, 2003). It is used directly by Boughton (2015) in estimating transmission losses in streams and McMahon and Nathan (2025) utilised  $K_{max}$  as a key to their procedure to estimate hydraulic properties of unconfined aquifers from daily streamflow data. We will include these additional references and commentary within Section 2.1.

- Line 86. What is the mean of 'ile'. Is it percentile? Response: Yes. We need to define the term '(percentile)' after the first use of 'ile.
- Figure 3. How is the starting point of the MRC ...? Response: This is specifically described in the last paragraph in Section 2.2 (Lines 101-107, original manuscript).
- 8. Figure 3. How ... is the observed recession superimposed on the calculated MRC? Response: Again, thank you, our explanation was not particularly clear. For this paper the superimposition of an observed recession on an MRC plot was simply achieved by translating the observed recession data horizontally along x-axis (representing time in days) until the observed recession overlaid one of the computed MRC curves that covered the continuum as shown in Figures 3-5. We will include an explanation of this in the first paragraph under Section 3.

**References associated with Anonymous Referee #1**

Boughton WC, 1995. Baseflow recessions. Aust. Civ. Eng. Trans. CE37, 9-13.
Brutsaert W, Lopez JP, 1998. Basin-scale geohydrologic drought flow features of riparian aquifers in the southern Great Plains. *Water Resour. Res.*, 34 (2), 233-240., https://doi:0043-1397/98/97WR-030685

- Brutsaert W, Nieber JL, 1977. Regionalized drought flow hydrographs from a mature glaciated plateau. *Water Resour. Res.* 13, 637–643 https://doi:7W0132.
- Chapman TG, 2003. Modelling stream recession flows. *Environ. Modell. Softw.*, 18, 683–692, https://doi:10.1016/S1364-8152(03)00070-7

Hall FR, 1968. Base-flow recessions—A review. *Water Resour. Res.*, 4, 973–983. Horton RE, 1933. The role of infiltration in the hydrologic cycle. *Trans. AGU*, 14, 446–460. McMahon TA, Nathan RJ, 2025. Estimating hydraulic properties and residence times of

- unconfined aquifers. *J. Hydrol.*, 654, 132861, doi.org/10.1016/j.jhydrol.2025.132861 Nathan RJ, McMahon TA. Evaluation of automated techniques for base flow and recession
- analyses. Water Resources Research, 26, 1465–1473, 1990.
- Rupp, D.E., Schmidt, J., Woods, R.A., Bidwell, V.J., 2009. Analytical assessment and parameter estimation of a low-dimensional groundwater model. *J. Hydrol.* 377, 143–154. https://doi.org/10.1016/j.jhydrol.2009.08.018.

Tallaksen LM, 1995. A review of baseflow recession analysis. J. Hydrol., 165, 349–370.

**Response to Anonymous Referee #2**

This technical note presents a concept that the master recession curve (MRC) is time-variable. While the manuscript introduces some interesting ideas, the ideas and their novelty are not fully described and developed to a degree that would warrant publication at this stage. The current manuscript is difficult to follow, and the ideas need to be better organized and streamlined. Thus, I recommend rejection.

Response: We appreciate the reviewer's time and effort in providing the comments below, and we expect that value of this contribution will be more apparent once we have updated the manuscript to address all reviewers' comments.

**Major Comments**

**1. Unclear Novelty and Lack of Litureature Review**

The novelty of the manuscript is unclear. Numerous previous studies have discussed the timevariability of flow recession dynamics. As the authors themselves note, 'an MRC represents the average rate of decline in streamflow,' implying that the MRC characterizes the representative behavior of a catchment. Therefore, it is not surprising that individual flow recession events may deviate from the MRC.

Response:

- 1. The novelty in our paper is represented in Figures 3-5. Firstly, no one has previously suggested that the master recession curve (MRC) is a continuum of curves rather than a single curve. Furthermore, the curves making up the continuum can be specified by an appropriate percentile value that is explicitly representative of natural variability. Secondly, Figures 3 5 are unique in that this is the first time that observed data are shown to represent a continuum of master recession curves. Moreover, the plotted observed data extend beyond the maximum recession constant and confirm the effect of stream evaporation and adjacent riparian evapotranspiration. Thirdly, the maximum recession constants that vary across the continuum bring consistency between the two procedures (hydrologic and hydraulic) that are used to estimate residence times. This is outlined in McMahon and Nathan (2025), a paper that had not been published when our draft was submitted for review.
- 2. Regarding the comment that "Numerous previous studies have discussed the timevariability of flow recession dynamics", we reviewed 59 papers since the classical Brutsaert and Nieber (1977) paper that deal with recession analysis. Of the 40 papers that discuss recession slope analysis (-dQ/dt ~ Q), only 6 discuss time variability and only one (Gao et al, 2023) explicitly characterises recession behaviour in probabilistic terms. This is a particularly valuable development as it provides the means to estimate differences in recession behaviour that result from differences in antecedent

conditions, thus avoiding the well-known limitation that MRCs only represent "average" conditions.

Uncertainty associated with the MRC has also been widely discussed in the literature. Response: In response to this query, we have reviewed 13 papers published since 1977 that deal explicitly with MRC, and except for our recent paper (McMahon and Nathan, 2025), no paper deals with the uncertainty in the MRC within the  $Q \sim t$  approach. As noted in the previous point, while previous papers recognise the limitations of ignoring variability in the MRC, our paper is the first to provide the means to quantify the associated uncertainty using a simple non-parametric technique.

**Furthermore, the authors appear to suggest that the MRC itself is time-variable; if this is the case, the definition must first be clarified—particularly since it is unclear whether it should still be referred to as a 'master' recession curve.**

Response: Yes, we do suggest the MRC is time-variable but have tagged the MRC with a prefix that is indicative of its probability of exceedance, for example, 10%'ile MRC, 30%'ile MRC, etc that addresses the concern of the reviewer. That approach allows the word 'master' to be included in the term, as this represents the expected differences in recession behaviour arising from differences in antecedent conditions.

**As previously noted, many studies have addressed the uncertainty related to the MRC, either directly or indirectly, within the broader literature on recession analysis, even if the term 'master recession curve' was not always explicitly used (e.g., Kirchner, 2009).**

Response: As we observed above, we do not agree with this observation. Kirchner (2009) does not address uncertainty in terms of time variability although he does consider sensitivity to change in storage and discusses the scatter between -dQ/dt and Q. Gao et al (2023) is one other paper that treats the scatter between -dQ/dt and Q in a probabilistic manner. Our method explains and quantifies the associated variability using the Q ~t approach and provides the means to estimate the resulting differences in recession behaviour.

**As indicated in Lines 209–212, Gao et al. (2023) considered the uncertainty (see also Thomas et al., 2015). Although a difference from Gao et al. (2023) is briefly mentioned, it is not sufficiently developed to convincingly establish the uniqueness of the current study.**

Response: The Gao et al. (2023) does introduce uncertainty into the MRC based on -dQ/dt ~ Q analysis. However, the method uses a theoretical parametric approach which yields recessions that are concave upwards, which is not representative of many streams, for example, in Australia (Boughton, 2015) or in the United States (Ye et al., 2014). We thus do not think their approach should be elaborated on in the more direct time-based method that we adopt, though we do note the probabilistic parallels in their approach. The primary goals of Thomas et al. (2015) were "... to evaluate numerous objective innovations for the characterization of hydrograph recessions and to show that the combination of these innovations can lead to an improved scientific understanding of the behavior of streamflow hydrograph recessions." (p. 110). The paper does not deal with uncertainty in terms of time variability.

The discussion of related work appears too late in the manuscript (e.g., Gao et al., 2023, in Lines 209–212) or is missing entirely (e.g., Thomas et al., 2015). A more thorough and critical literature review should have been presented in the Introduction. The current review in Lines 27–44 is too brief and merely lists previous studies without offering sufficient critical analysis. A deeper review—incorporating Gao et al. and other relevant studies addressing similar aspects—should be integrated earlier in the manuscript to better contextualize the research.

Thomas, B.F., Vogel, R.M., Famiglietti, J.S., 2015. Objective hydrograph baseflow recession analysis. J. Hydrol. 525, 102–112.

Response: As we noted in response to the first reviewer, if offered an opportunity to resubmit we will expand and recast the tabular material into text format with some concluding comments. We will elaborate on the probabilistic parallels with Gao et al. (2023) but do not see the appropriateness of including mention of the Thomas et al. (2015) paper.

Additionally, accounting for losses in flow recession dynamics is not a novel concept; see, for example, Wang and Cai (2009).

Wang, D.; Cai, X. (2009) Detecting Human Interferences to Low Flows through Base Flow Recession Analysis. Water Resources Research

Response: It is unclear what the reviewer is referring to. The paper only addresses transmission loss as a fact. We do not carry out any specific analysis nor comment about it being a novel concept, though we note here that one of the benefits of our non-parametric approach is that it properly accounts for such losses where they do occur.

In the discussion, the authors employ a two-bucket model, which is commonly used in the flow recession literature; however, relevant previous studies are not referenced. See Gao et al. (2017) for example.

Gao, M.; Chen, X.; Liu, J.; Zhang, Z.; Cheng, Q. (2017) Using Two Parallel Linear Reservoirs to Express Multiple Relations of Power-Law Recession Curve. Journal of Hydrologic Engineering.

Response: Thank you. A comment will be added in Section 4.2 and references.

**2. Regarding the transmission loss**

The authors appear to emphasize transmission loss; however, it is unclear whether transmission loss can be adequately assessed using discharge data alone, based on the reasoning provided in the manuscript.

Response: As noted earlier we include transmission loss in our discussion as it is a physical process that influences the behaviour of the observed recessions.

In Lines 60–62, the authors argue that transmission loss is reflected in the deviation of the MRC (which can be estimated from the data) from the recession behavior the catchment would exhibit in the absence of transmission loss—though there is no guarantee that such a reference curve can be reliably determined from the data.

Response: Boughton (2015) used only daily discharge data to estimate the transmission loss for 100 streams located in the temperate-humid east coast of Australia. Our paper 'Baseflow and transmission loss: A review' (McMahon and Nathan, 2021) provides a detailed discussion of transmission loss. We will add this reference as an appropriate point in a revised manuscript.

Throughout the manuscript, it also seems to be implicitly assumed that a catchment without transmission loss would necessarily follow linear reservoir behavior, which requires further clarification and justification.

Response: When the recession reached the maximum recession constant, and there is no transmission loss, the recession will follow linear reservoir behaviour. This assumption has been the basis of baseflow recession since recession analysis was first proposed by Boussinesq in 1903 and, independently, by Maillet (1905). Horton (1933), Horner and Flint (1936), and Barnes (1939) followed. We will add these references to better place this assumption in the context of previous literature.

Furthermore, the assumption that a faster flow recession than exponential decay (as predicted by linear reservoir theory) is necessarily attributable to transmission loss is not well justified. Is this merely an assumption? Are there no other possible explanations for observing a steeper decline in discharge besides transmission loss?

Response: There may be other explanations in special circumstances but we are unaware of any. Various authors have attributed the influence of transmission losses, mainly as evaporation/riparian evapotranspiration, as a cause of the steeper slope than exponential. Examples for this include Stewart and Boughton (1983), McMahon and Finlayson (2003), Rupp and Woods (2008, p 2669). Ye et al. (2014) found increases in slope to be related to increases in aridity index.

Moreover, the emphasis on transmission loss appears to diminish as the manuscript progresses, with greater focus shifting toward the role of initial conditions at the onset of flow recession. Although the authors seem to attempt to link these two aspects, the connection is not fully developed and is difficult to follow (e.g., Lines 261–263).

Response: Our method does not depend on the conflation of these two factors as they merely represent different physical processes which help to explain the aleatory uncertainty involved in recession behaviour. The introduction of transmission loss as stream evaporation plus riparian evapotranspiration is noted as appropriate throughout the manuscript to complete our explanation of the recession process taking place. We will check to ensure our phrasing does not imply that one of these processes is more or less important than the other.

**3. Method**

**The method introduces certain criteria and parameters without providing sufficient justification or clear explanation. Some examples are outlined below.**

Response: After reading both reviewers comments, we acknowledge out methodology needs to be more clearly set out. The changes we propose are:

- The last paragraph in the Introduction will be revised to make clearer that we are dealing with time-based analysis (Q ~ t) rather than the time-derivative-based analysis (-dQ/dt ~ Q).
- 2. A new paragraph added immediately following the heading "2 Adopted approach" introducing our two-step analysis.
- 3. The section outlining correlation procedure to be expanded.
- 4. The construction of MRC to be slightly expanded.
- 5. Heading 2.2 to be replaced with 'Superimposing observed recessions on MRCs'
- 6. Section 3 heading "Results" to be replaced by 'Application'.
- Criterion 2 (Line 117) It appears that the authors exclude all recession events that do
  not exhibit a 'maximum recession constant' midway through the recession.
  Response: Our sentence is unclear. Given an opportunity we will delete ", prior to the
  maximum recession constant...". For a continuous concave upward recession, there is
  no maximum recession constant.
- It seems that the authors have chosen to focus on a specific subset of events, but this decision is not well explained in the manuscript. In addition, this criterion could potentially exclude a large number of recession events across many catchments. Response: We are not clear what "subset of events" the reviewer is referring to here. In line with all previous studies, we simply chose events that are unaffected by rainfall over a specified minimum period; as discussed in Section 2.2, we adopted a minimum period of seven days, which is representative of the range adopted by other authors.

• Therefore, it should be introduced and justified much more carefully. Also, this criterion appears to implicitly rely on the linear reservoir theory. It is unclear why such a model should be adopted in this context. Response: This is incorrect. We make no assumptions about the degree of linearity in the storage reservoir as we use a non-parametric approach that is based on observed

the storage reservoir as we use a non-parametric approach that is based on observed streamflow behaviour. In selecting the observed recessions to be superimposed on the MRCs (e.g. Figure 3), the recessions are chosen as set out in Section 2.2. We will carefully review our descriptions of the method to ensure that the reader is not left with this misconception.

- The various criteria for selecting recession periods are scattered across the manuscript and should be consolidated and explained earlier. Response: We disagree with this observation, as the only place where we are specific about the selection criteria adopted is in Section 2.2. But again, we will carefully review our text to ensure that the reader is not left with this impression.
- Several choices—such as excluding periods where dQ/dt = 0 (Lines 127–128) and applying a minimum duration threshold—are not adequately justified. For instance, by excluding periods with dQ/dt = 0, the authors may unintentionally bias the analysis toward steeper portions of the recession, particularly when flows are low and measurement resolution becomes an important consideration.

Response: The criteria adopted are similar to those used in all the relevant literature:

- "Regarding dQ/dt = 0 (Lines 127-128)". If two consecutive discharges are equal, then, by definition, recession ceases. This is a reasonable guideline to adopt in our study and one adopted by most investigators studying recessions using Q~t or in -dQ/dq~Q methodologies. (We are aware that some analysts not only include -dQ/dt = 0 but also values <0, e.g., Kirchner (2009))</li>
- "minimum duration period". As noted in Lines128-130, adopting seven consecutive recession data points is a trade-off. Other have used 12 days (Hameed et al., 2023),10 days (van Dyke, 2010), 7 days (Whitaker et al., 2022), 6 days (Gao et al., 2023) and 5 days (Parra et al., 2023).
- The number of selected events are too small; it is unclear whether meaningful statistical analysis is possible (Table 1).
   Response: As reported in Table 1, the number of plotted recessions is 22, 55, 64 and 30.
   These are certainly of sufficient size to carry out a non-parametric analysis and at least three of four are of sufficient size to complete a parametric analysis.

Additionally, the method is not described clearly enough for readers to follow easily. See below for some examples.

- The correlation technique is not sufficiently explained; the reader should not be required to consult external sources. Response: We accept the need to provide a more detailed explanation. This will be done in a revised manuscript within Section 2.1.
- Manual allocation (L163–170) is poorly described and hard to follow. Response: Actually, the manual allocation is not described in Lines 163-170, but we do agree that we need to be clearer about this. We propose to amend the text in Section 3 to provide the required clarity.
- It is unclear how starting points in Figure 3 were chosen.

Response: The starting conditions are described in Lines 103-105, but we will revisit this text to provide more clarity.

• Some figure descriptions conflict with the figure contents (e.g., general wet/dry periods in Figure 4).

Response: The caption in Figure 4 will be amended to note that the data are drawn separately from the wet 6-months and dry 6-months periods. We will carefully check other captions to ensure that this is not the case elsewhere.

**Minor Comments**

• **L46–53**: This paragraph appears intended to describe applications, but the content within the parentheses following Nathan and McMahon, and O'Brien, discusses methods or techniques rather than applications.

Response: Thank you. We will check the text to ensure the content within parentheses is correct. In the case of Nathan and McMahon (1990), we should have written "(applying MRCs to 186 Australian catchments)" under application and O'Brien et al. (2014) should have been listed in the previous paragraph.

• L51: It does not seem appropriate to discuss "residence time" based solely on recession analysis.

Response:

- 1. L51 refers to a paper Raeisi (2015) who used the MRC to estimate residence time.
- 2. Residence time is not referred to anywhere else in the manuscript. While we agree residence time can be estimated by other methods, given our focus it is inappropriate to include any discussion of the topic here. We provide an extensive discussion of the so-called hydrologic and hydraulic methods for estimating residence time in McMahon and Nathan (2025).
- L62: Is transmission loss the only possible explanation for the observed deviation? On what basis is this claim made? Is it an assumption? Response: It is conceivable that evapotranspiration from the discharging aquifer due to

deep rooted vegetation could result in this effect. Once a recession reaches B (the maximum recession constant) (Figure 1), aquifer discharge is from a single storage. Boughton (2015) utilised this concept to estimate transmission losses from 100 streams located in the temperate-humid east coast of Australia. However, as noted above, our approach is based on the analysis of observed data and we do not rely on making any assumptions around the physical processes which contribute to the natural variability of recession behaviour.

- **L66**: The reference to "the above limits" is unclear—what limits are being referred to? Response: "the above limits" is a poor choice of phrase. We are referring to the constraint 0<Kj<1. This section will be rewritten and incorporated into a more detailed discussion of the correlation method suggested by Reviewer 1.
- **L67**: This step seems to be clearly explained within this manuscript rather than referring readers to another study.

Response: The reference to Linsley et al. (1982) refers to the number of days daily discharges are eliminated to ensure analysis is based on discharges unaffected by rainfall. This will be made clearer in a revised manuscript.

- L75: The term "traditionally" is used multiple times, but it is unclear what specific tradition is being referenced. There are various interpretations of the master recession curve (MRC) in the literature. The authors should clearly define the version of the MRC they are using rather than relying on the vague notion of a traditional definition. Response: We will delete the word "traditionally". Although the term 'master recession curve' was not introduced into the hydrology literature until 1964 by Toebes and Strang (1964), it was initially called a normal depletion curve by Horton (1933), groundwater depletion curve by Grundy (1951) and a composite recession curve by Linsley et al. (1958). It is defined by a plot of discharge (usually daily) against time where discharge is in log10 units. We will add a paragraph based on this explanation.
- **L84**: The meaning of "sub-storages" is unclear. Response: We will replace this term with 'aquifers'.
- L92: The phrase "stochastic process rate" is ambiguous. Response: We propose to replace the phrase "who described an MRC in terms of a stochastic process rate" with 'who analysed the structure of the recession curve within a stochastic rather than in a deterministic framework'.
- L164: Clarify what is meant by "the value of the observed daily recession." Response: We will replace "value" with 'magnitude'.
- **L193**: What is meant by "the minimum that can be estimated under field conditions"? Response: We will replace "estimated" with 'measured'.
- L210–212: The discussion of previous results is not well handled, and the rationale for the authors' interpretation is not sufficiently explained. Response: This sentence refers to Gao et al. (2023) paper. The last sentence is confusing. We will delete it.

 L216: The statement here is very unclear and needs clarification. Response: We understand your difficulty here. We agree the sentence in Lines 215 -217 is ambiguous. Many studies involving recession analysis (Q ~ t) that produce Q ~ t diagrams do not extend the time scale much beyond the maximum recession constant, presumably in the belief that the curve continues in a concave manner. However, our Figure 3, for example, shows that observed daily recessions can continue beyond the maximum recession constant – see plotted values superimposed on the 10, 30 and 50%'ile MRCs. In a revised manuscript, we will rephrase this observation to make the paragraph including Lines 215-217 clearer.

- L222–228: Either Figure 4 or the corresponding description seems to be inaccurate. It is unclear what is meant by "general wet and dry periods." Response: There is a paragraph beginning at Line 172 explaining Figure 4. We propose to make several minor changes:
  - 1. Line 172 delete "with reference to Fig. 3"
  - 2. Amend the Figure 4 caption to 'Comparison of observed daily recessions for data drawn separately from the wet 6-months and dry 6-months periods superimposed on the estimated MRCs for Gibbo River at Gibbo Park (401217'.

• L227: It is questionable whether the phrase "near identical" is necessary or helpful in this context.

Response: We agree. We will delete this sentence.

- L231-232 "initial climate condition": I guess it is not "initial climate condition". • Response: Thank you. We will replace "initial climate condition" with 'initial storage in the aquifer'.
- Definitions of terms such as "aleatory uncertainty" and "epistemic uncertainty" are • unclear or appear to be misused.

Response: We will add additional text to the Introduction that defines these terms, and will include a reference to the McMahon and Nathan (2025) paper that describes the governing aleatory concepts in more detail.

**References associated with Anonymous Referee #2**

Barnes, B. S. (1939). The structure of discharge-recession curves. Transactions AGU, 20, 721-725.

- Grundy, F. (1951). The ground-water depletion curve, its construction and uses. Assemblee Gen. De Bruxelles. International Association of Hydrological Sciences, 2, 213–217.
- Hameed, M., Nayak, M.A., Ahanger, M.A., 2023. Event-based recession analysis for estimation of basin-wide characteristic drainage timescale and groundwater storage trends. Water Resour. Res. 59, e2023WR035829. https://doi.org/10.1029/ 2023WR035829.
- Horner, W. W., & Flynt, F. L. (1936). Relation between rainfall and runoff from small urban areas. Transactions ASCE, 1926, 141–183
- Horton, R. E. (1933). The role of infiltration in the hydrologic cycle. Transactions AGU, 14, 446-460.
- Maillet, E. (1905). Essai d'hydraulique souterraine et fluviale: Librairie scientifique. Paris: A. Hermann.
- McMahon TA, Finlayson BL, 2003. Droughts and anti-droughts: the low flow hydrology of Australian rivers. *Freshwater Biology* 48, 1147-1160.
- McMahon TA, Nathan RJ, 2021. Baseflow and transmission loss: A review. WIREs Water Wiley https://doi.org/10.1002/wat2.1527
- McMahon TA, Nathan RJ, 2025. Estimating hydraulic properties and residence times of unconfined aquifers. J. Hydrol., 654, 132861, doi.org/10.1016/j.jhydrol.2025.132861
- Nathan RJ, McMahon TA, 1990. Evaluation of automated techniques for base flow and recession analyses. Water Resour. Res. 26 (7), 1465-1473. https://doi.org/10.1029/
- Parra, V., Munoz, E., Arumí, J.L., Medina, Y., 2023. Analysis of the behavior of groundwater storage systems at different time scales in basins of South Central Chile: A study based on flow recession records. Water 2023 (15), 2503. https://doi.org/ 10.3390/w1514250.
- Rupp, D. E., & Woods, R. A. (2008). Increased flexibility in base flow modelling using a power law transmissivity profile. Hydrological Processes, 22, 2667–2671
- Stewart, B., & Boughton, W. C. (1983). Transmission loss in natural streambeds—A review. Institution of Engineers Australia, National Conference Publication, 83(13), 226–230.
- Van Dijk, A.I.J.M., 2010. Climate and terrain factors explaining streamflow response and recession in Australian catchments. Hydrol. Earth Syst. Sci. 14, 159–169 https:// www.hydrol-earth-syst-sci.net/14/159/2010.
- Whitaker, A.C., Chapasa, S.N., Sagras, C., Theogene, U., Veremu, R., Sugiyama, H., 2022. Estimation of base flow recession constant and regression of low flow indices in eastern Japan. Hydrolog. Sci. J., 67 (2), 191-204, https://doi.org/10.1080/02626667.2021.2003368.

Ye, S., Li, H. Y., Huang, M., Ali, M., Leng, G., Leung, L. R., Wang, S. W., & Sivapalan, M. (2014). Regionalization of subsurface stormflow parameters of hydrologic models: Derivation from regional analysis of streamflow recession curves. *Journal of Hydrology*, 519, 670–682.